# Alcohol Consumption and Mild Cognitive Impairment: A Mendelian Randomization Study from Rural China

**DOI:** 10.3390/nu14173596

**Published:** 2022-08-31

**Authors:** Yi Cui, Wei Si, Chen Zhu, Qiran Zhao

**Affiliations:** 1College of Economics and Management, China Agricultural University, Beijing 100083, China; 2Academy of Global Food Economics and Policy, China Agricultural University, Beijing 100083, China

**Keywords:** alcohol consumption, mild cognitive impairment, Mendelian randomization, rural China

## Abstract

Alcohol consumption has been associated with the risk of mild cognitive impairment (MCI) in observational studies. The result is inconsistent and whether the association is causal remains unknown. To examine the causal effect of alcohol consumption on MCI in rural China, this study used a cross-sectional dataset that included 1966 observations collected in rural China, of which 235 observations’ genotyping were collected. All participants accepted the MCI evaluation using Mini-Cog and were asked about the participants’ alcohol consumption behavior. The causal effect of alcohol consumption on MCI was investigated by Mendelian randomization (MR) of genetic variation in the aldehyde dehydrogenase 2 (*ALDH2* rs671) gene. The risk of MCI in Chinese rural areas was 43%. Alcohol consumption was causally associated with a higher risk of MCI under MR design. Parameter estimates of drinking or not (*b* = 0.271, *p* = 0.007, 95% CI = 0.073 to 0.469), drinking frequency during the past 30 days (*b* = 0.016, *p* = 0.003, 95% CI = 0.005 to 0.027), and the weekly ethanol consumption (*b* = 0.132, *p* = 0.004, 95% CI = 0.042 to 0.223) were all positive and statistically significant at the 5% level. In conclusion, there was a high risk of MCI in rural China, and alcohol consumption was causally associated with a higher risk of MCI.

## 1. Introduction

Mild cognitive impairment (MCI) is a preclinical transitional stage between healthy aging and dementia [1]. MCI means various disorders of the mind or intellectual activity, such as sensation, perception, memory, language, etc. It is a sign of brain dysfunction and is also one manifestation of normal brain aging [2,3,4]. Moreover, MCI is believed to precede dementia as an earlier state and a population with MCI will progress to dementia faster than age-matched healthy controls [5,6]. It is very important to prevent and treat chronic aging diseases such as MCI.

The prevalence rate of MCI reported varies greatly. Studies from America, Australia, Bulgaria, Mexico, and Japan have reported the prevalence rate of MCI ranges from 6.5% to 39.1% [7,8,9,10,11,12]. In the case of China, the study of MCI has been taken in some large cities such as Shanghai and Beijing, which found the prevalence rate of MCI ranges from 2.4% and 35.9% [6]. With the rapid increase in the number of elderly people in rural China, age-related diseases are becoming more common [13]. The rural population accounts for one-third of China’s population and their health is worse than urban people. However, there are few studies in Chinese rural areas [14]. It should be concerned with the prevalence of MCI in Chinese rural areas.

Many studies have analyzed risk factors for MCI, such as age, gender, educational level, and cardiovascular disease [15,16,17,18,19]. Alcohol consumption is considered a possible risk factor for MCI [20]. China is a country with a large production and consumption of alcohol. From World Health Organization (WHO) report, per capita, alcohol consumption in China has been increasing year by year, with an increase of 76%: 4.1 L in 2005, 7.1 L in 2010, and 7.2 L in 2016 [21]. The alcohol consumption in rural was nearly 1.4 times higher than in urban areas [22]. Moreover, alcohol abuse and dependence are common disorders in Chinese rural areas [23]. Existing research on the relationship between alcohol consumption and MCI remains controversial. A cohort study revealed participants who drank alcohol regularly in middle age were more likely to have MCI [24]. Another study indicated light–moderate alcohol drinking decreased risks for dementia in elderly patients with MCI [25]. Other studies found no significant association between alcohol consumption and the incidence rate of MCI [26,27]. As far as we know, there are few studies on the impact of alcohol consumption on MCI in China, especially in rural China.

It is difficult to estimate the causal impact of alcohol consumption and MCI using observational data. The observed MCI effects may have reverse causality, meaning that individuals with MCI may deliberately control alcohol intake considering their health condition [25]. Furthermore, the observed effects may be caused by some confounding factors, such as socio-economic status, dietary habits, or other health-related behaviors [22]. It is of great significance to study the causal relationship between alcohol consumption and MCI for evaluating the benefits or hazards of alcohol consumption.

Our study aims to analyze the causal impact of alcohol consumption on the risk of MCI. We investigate the causal effect of alcohol consumption on MCI by using Mendelian randomization (MR) on the genetic variation of the aldehyde dehydrogenase 2 gene (ALDH2 rs671). ALDH2 rs671 is shown to have the strongest association with alcohol consumption [28], and we demonstrated that ALDH2 was not associated with the risk of MCI. This paper is the first to estimate the causal effects of alcohol drinking on the resulting MCI, at least in the context of rural China.

## 2. Materials and Methods

### 2.1. Sample Collection

This study uses a cross-sectional dataset which collected in rural areas of Xinjiang, Shandong, Henan, Anhui and Heilongjiang provinces in 2019. Using an income-stratified sampling method, 33 counties in total were randomly selected based on per capita gross value of industrial output in each county. In each selected county, we randomly selected 6 villages. The sample selection procedure is shown in Figure 1. In each village, we randomly enrolled 10 famers and there is a total of 1975 observations. Before the survey begun, trained investigators explained the purpose and content of the survey to each participant in detail and told them that the survey was completely voluntary and that all information would be kept confidential. If they had any questions about the survey, they could contact the research team at any time. All participants signed an informed consent form. The Institutional Review Board of China Agricultural University approved the protocol (Protocol ID: CAUHR-2020-005).

The survey collected regular demographic and socioeconomic status information of participants, such as age, gender, education level, annual household income, as well as personal alcohol consumption information of participants. In addition, 1 mL saliva samples were collected for genotyping in professional test tubes from part of respondents. A total of 1966 observations were collected, 235 of which were genotyped, excluding individuals who did not complete the questionnaire and whose quality was not qualified.

### 2.2. Genotyping

DNA was extracted from saliva samples using the Illumina WeGene V2 Array (Shenzhen, China). Imputation and quality control were performed using PLINK (1.90 Beta), SHAPEIT (v2.17), and IMPUTE2 (v2.3.1).

### 2.3. Measure of Alcohol Consumption

We investigated alcohol consumption behavior in three aspects. First, we conducted a binary measurement of whether the respondents drank or not, where 0 represented current non-drinkers and 1 represented current drinkers. Second, we asked participants about the total frequency of consumption of three kinds of Chinese common alcohol (beer, wine, and liquor) during the past 30 days. Third, using the three information on the consumption frequency of three Chinese common alcohol (beer, wine, and liquor) for one month, the average amount of the three alcoholic drinks each time, and the alcohol content of each alcoholic drink, we calculated each respondent’s weekly ethanol (100 g of pure alcohol per week) consumption as a continuous measure of alcohol consumption.

### 2.4. Genetic Instruments

MR is an ingenious causal research design, which uses genes as instrumental variables [29,30]. The MR approach relies on random assignment of genes during meiosis in humans, resembling the random assignment into treatment groups in randomized controlled trials (RCT) [23,31,32,33,34]. In MR studies of alcohol use, the aldehyde dehydrogenase 2 gene (ALDH2 rs671) is usually used. This gene encodes enzymes involved in ethanol metabolism pathway, which can change the metabolic balance of acetaldehyde in human body [28]. Ethanol is first converted to acetaldehyde by alcohol dehydrogenase (ADH), and then further converted to acetate by acetaldehyde dehydrogenase (ALDH).

The number of effect alleles in ALDH2 rs671, specifically the number of the A allele, largely determined the enzyme activity of ALDH. The three genotypes for ALDH2 rs671 allele in East Asian populations are GG (the number of A allele is 0), AG (the number of A allele is 1), and AA (the number of A allele is 2). Carrying the A allele significantly decreases the detoxification of acetaldehyde generated during alcohol metabolism in human body [28,35], making people feel uncomforted after drinking alcohol.

### 2.5. Measure of MCI

All participants accepted mild cognitive impairment evaluation using Mini-Cog [36]. Mini-Cog combined the clock drawing task (CDT) (normal versus abnormal) and three words memory (0–3 scores) into a tool to assess MCI. These tests did not require high education level of the respondents and were easily understood by the respondents. Whether the participant has MCI or not is determined based on the completion of CDT and three-word recall. The specific simple decision rules are shown in Figure 2 [36]. Specifically, there were three rules: subjects recalling none of the words were classed as MCI; those recalling all three words were classed as non-MCI; and those with intermediate word recall (1–2) were classed based on the CDT (abnormal = MCI, normal = non-MCI).

### 2.6. Statistical Analysis

We used multivariable linear regression to examine the correlation between alcohol consumption and MCI. Next, we verified the effectiveness of genetic instrumental variable (ALDH2), and then conducted MR analysis using two-stage least squares (2SLS) to evaluate the causal relationship between alcohol consumption and MCI. The concept of MR is analogous to randomized controlled trials (RCT) which are difficult to implement but can help us make causal inferences about the effects of alcohol use/drinking behavior on MCI [33,34]. Figure 3 illustrates the design of MR in our study. Considering that demographic characteristics and socioeconomic status factors might be highly correlated with alcohol use and MCI, we adjusted for age, gender, ethnic minorities, income, education years and province fixed effects in all regressions. In MR analysis, we adjusted for the number of parents who drink alcohol to deal with potential threats from dynastic effect [37].

All statistical analyses were conducted using Stata version 14.0 (StataCorp LP, College Station, TX, USA). Significance was set at *p* < 0.05.

## 3. Results

As shown in Table 1, the average age of the respondents in the genetic sample was 49.7 years old, with 8.6 years of educational experience. There were 183 males in the sample, accounting for 77.87%. A total of 51.9% of the sample were current drinkers and 48.1% were current non-drinkers. The mean total frequency of three kinds of Chinese common alcohol (beer, wine, and liquor) during the past 30 days was 9.6 (drink about once every 4 days on average). The average weekly ethanol consumption was 120 g. We found that the risk of MCI in the sample area was 43%. About the genotype of ALDH2 rs671, 34.5% of respondents are A-allele carriers (genotypes of AA and AG). Specifically, the percentages of genotype AA and AG are 2.6% and 31.9%.

Table 2 reported estimates of alcohol consumption on MCI from multivariable linear regressions and MR analyses on the genetic sample. From Table 2, we found none of the key explanatory variables were significantly associated with the risk of MCI, possibly because alcohol consumption could still be confounded by various unobserved factors, such as socio-economic class, and drinking attitude.

*ALDH2* rs671 is a valid genetic instrument variable that relied on the assumption of relevance, independence, and exclusion restriction. Firstly, we confirmed that the correlations between ALDH2 rs671 and alcohol consumption existed stably in the sample, regardless of adjustments for additional controls. Further R-squared results suggested that *ALDH2* rs671 could explain 3.6–16.8% of the total phenotypic variation in different alcohol consumption measurements, suggesting *ALDH2* rs671 was a valid genetic instrument variable. Secondly, the first-stage F statistics of MR model were used for weak instruments test. We found the first-stage F statistics for different models exceeded the conventional cut-off of 10 for weak instruments, which means the genetic instrument variable was a strong instrument in our MR design. Thirdly, through the query in phenoscanner (V2) (a database of human genotype–phenotype associations), there were no direct links of *ALDH2* rs671 with MCI-related phenotypes, further supporting the validity of this genetic instrument variable.

Table 2 reported MR results by *ALDH2* rs671 as an instrumental variable in the genetic sample. In the MR analysis, three alcohol consumption variables were causally associated with a higher risk of MCI. Specifically, parameter estimates for current drinking or not (*b* = 0.271, *p* = 0.007, 95% CI = 0.073 to 0.469), drinking frequency during the past 30 days (*b* = 0.016, *p* = 0.003, 95% CI = 0.005 to 0.027), and the weekly ethanol consumption (*b* = 0.132, *p* = 0.004, 95% CI = 0.042 to 0.223) were positive and statistically significant at the 5% level.

The current never-drinking group included previous drinkers who do not drink alcohol now. Previous drinkers may stop drinking because of underlying health problems. Considering that, we generated an alcohol abstinence years variable to further distinguish between the never-drinking sample and the sample who are non-current drinkers but previous drinkers. We dropped the sample who had abstained from alcohol in the past five years from the never-drinkers (*n* = 15) and ran an MR analysis. From Table 3, the MR regression results excluding the recent five years of abstinence are basically consistent with the regression results of the total genetic sample. As can be seen in Table 1, most of the participants were male, and we analyzed MR regressions for the male sample. From Table 3, the MR regression results of the male sample are basically consistent with the regression results of the total genetic sample.

## 4. Discussion

We found the risk of MCI in rural China was 43%. Xue, Li, Liang and Chen [14] found that prevalence of MCI in the Chinese elderly population was 14.71%. Nie, Xu, Liu, Zhang, Lei, Hui, Zhang and Wu [6] found prevalence of MCI for the elderly population was 12.7%. The reported prevalence of MCI difference from our finding may be due to different assessment procedures and sample regions. The samples in our study were all from Chinese rural areas, while the samples in Xue, Li, Liang and Chen [14] and Nie, Xu, Liu, Zhang, Lei, Hui, Zhang and Wu [6] were mostly from Chinese urban areas. The prevalence of MCI in rural areas is generally higher than that in urban areas, because of difference life environments, medical conditions, and education levels between urban and rural areas. As more and more young and middle-aged rural residents worked in cities, there is a growing demand for health care, especially about cognitive impairment, for older people who stay in Chinese rural regions. Our findings provide new evidence for the need and urgency of monitoring cognitive function in rural China.

This study is one of the first studies using an MR design to investigate the causal relationship between alcohol consumption and MCI in the Chinese rural population. Using an MR analysis of 235 participants from Chinese rural areas, we found that the effect of alcohol consumption on the higher risk of MCI was causal. This finding was consistent with MR research that examined the causal relationship between alcohol consumption and Alzheimer’s disease, which observed people who drank 1 SD (1.90 drinks/week) of alcohol were twice as likely to develop Alzheimer’s disease compared with normal people at a given point of time [38]. There are other studies that are consistent with our findings. Järvenpää et al. [39] in a longitudinal study on 554 twins persons above for 25 years, found drinking exceeding the amount of five bottles of beer or one bottle of wine on one occasion at least monthly was associated with a relative risk of 3.2 for dementia. Cherbuin et al. [40] found harmful alcohol consumption behavior (>42 units per week) increased risk of transitioning to MCI by analyzing 2082 subjects aged 60 to 64. Zhou, Deng, Li, Wang, Zhang and He [20] suggested that alcohol consumption was closely associated with cognitive impairment (*p* = 0.025). Although these studies are consistent with the conclusions of our finding, our research found that alcohol consumption increases the risk of MCI regardless of heavy drinking or not. Comparing with previous studies, our research focuses on Chinese rural residents and provides empirical analysis results of a non-elderly sample (average age of the sample is 49.7 years).

However, our findings contradict several studies that suggest that mild to moderate alcohol consumption can prevent MCI or dementia [24,25,41]. These findings were in correlational studies that were limited by selection bias, and the heterogeneous nature of the non-drinker comparison group [38]. Because of the random assignment of genotypes from parents to children, MR analysis reduces the effect of confounding factors and reverse causality, which is the same as the principle of RCT. The advantages of MR analysis provide stronger support for causal effect analysis. A Mendelian randomized study of older Australian men found that alcohol consumption was not a direct cause of cognitive impairment [42]. The sample of this study was the elderly male group over 65 years old and the tracked sample was healthier than the untracked sample, which may affect the results. Our study provides support for the cautious interpretation of casual effect of alcohol consumption on MCI, and further highlights the need for future studies to consider potential confounding factors.

The mechanism between alcohol consumption and MCI is under unclear. One mechanism for this result might be acute alcohol intoxication, which has been shown to interfere with memory and learning processes in animal studies [43,44]. Another possible pathway is that alcohol consumption has a high risk of head injury, which is a risk factor for dementia in adulthood [45].

There are several limitations in this study needed to be further addressed in later studies. The sample size in this study was relatively small, and there was a higher proportion of males in the sample. Chinese women rarely drink alcohol on a regular basis, and the large proportion of men in the sample may reduce the effectiveness of generalizing the results to women. In addition, there may be a time lag that exists between alcohol consumption and subsequent cognitive changes. There may be limitations in using alcohol consumption over the past 30 days to assess the relationship between alcohol consumption and cognitive change.

## 5. Conclusions

In conclusion, we found that the risk of MCI in Chinese rural area was 43%. MR analysis found that alcohol consumption was causally associated with a higher risk of MCI. That means that any amount of alcohol increases the risk of MCI. We believed that our study provides empirical reference and evidence for the development of comprehensive and targeted programs, to reduce the incidence of cognitive impairment by reducing alcohol consumption among Chinese rural residents. Nevertheless, further investigations are needed to clarify the potential underlying mechanisms.

## Figures and Tables

**Figure 1 nutrients-14-03596-f001:**
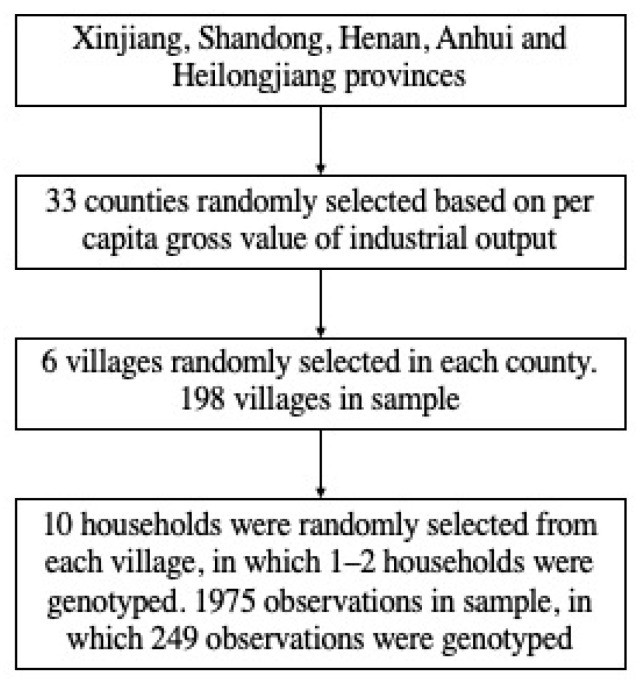
Diagram of household sample selection procedure.

**Figure 2 nutrients-14-03596-f002:**
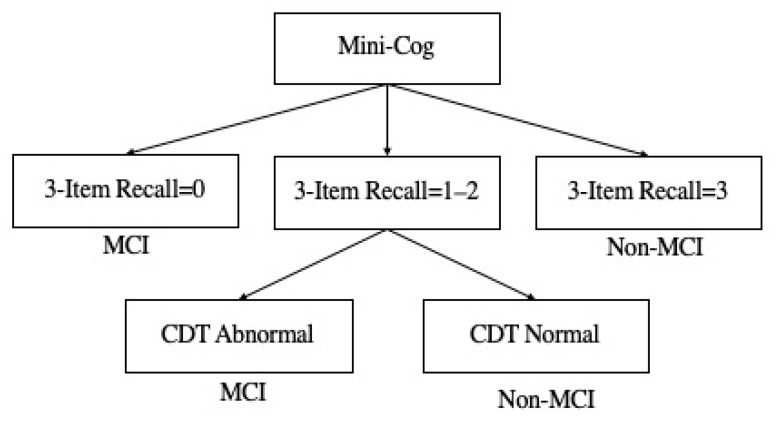
Mini-Cog scoring algorithm.

**Figure 3 nutrients-14-03596-f003:**
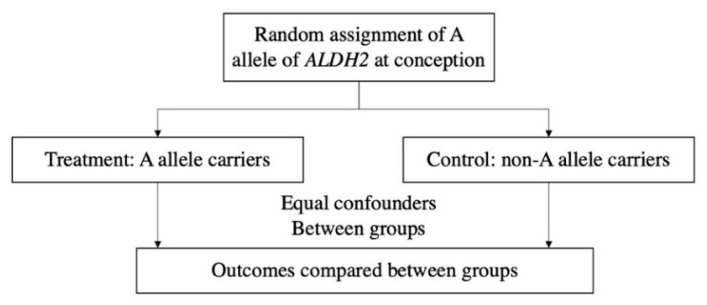
Design of Mendelian randomization in the current study.

**Table 1 nutrients-14-03596-t001:** Basic characteristics of genetic sample.

Variable	Definition	Observation	Mean	S.D.
Socio-demographic characteristics:				
Male	dummy; 0 = female; 1 = male	235	0.8	0.4
Age	age measure by year	235	49.7	11.3
Ethnic minority	dummy; 0 = no; 1 = yes	235	0.1	0.3
Education	educational years	235	8.6	3.1
Income	household income is taking log	235	1.0	2.2
Parents drinker	the number of drinking parents	235	0.4	0.6
Drinking behaviors:				
Drinking or not	dummy; 0 = no; 1 = yes	235	0.5	0.5
Drinking frequency	drinking frequency during the past 30 days	235	9.6	16.7
Weekly ethanol consumption	100 g/week	235	1.2	2.4
The risk of MCI:				
MCI	dummy; 0 = no; 1 = yes	235	0.4	0.5
Genetic variants:				
ALDH2 (rs671)	AA/AG	81	34.5%	-
	GG	154	65.5%	-

**Table 2 nutrients-14-03596-t002:** Effects of alcohol consumption on MCI-OLS and 2SLS estimation results.

Variable	Multivariable Linear Regressions	Mendelian Randomization
(1) Drinking or not		
*b* (95% CI)	−0.045 (−0.333, 0.243)	0.271 (0.073, 0.469)
*p*	0.689	0.007
*F* (p)		34.88 (0.000)
(2) Drinking frequency during the past 30 days		
*b* (95% CI)	−0.000 (−0.012, 0.011)	0.016 (0.005, 0.027)
*p*	0.915	0.003
*F* (p)		37.23 (0.004)
(3) Weekly ethanol consumption (100 g/week)		
*b* (95% CI)	−0.006 (−0.054, 0.042)	0.132 (0.042, 0.223)
*p*	0.751	0.004
*F* (p)		31.89 (0.005)

MCI was defined as Mini-Cog [36]. Abbreviations: 95% CI represents 95% confidence interval. All regressions were adjusted for age, gender, ethnic minority, years of education, logarithm of household income, and province-fixed effects. In particular, the MR model adjusted for the number of respondents’ parents who drink alcohol. The first-stage Cragg–Donald F statistics (with values of *p*) is test statistics of the weak instrument and overidentification tests.

**Table 3 nutrients-14-03596-t003:** Sensitivity analysis of alcohol consumption on MCI–2SLS estimation results.

Variable	Drop Last Five Years of Abstinence Sample	Male Sample
Observations	221	183
(1) Drinking or not		
*b* (95% CI)	0.193 (0.024, 0.363)	0.243 (0.017, 0.469)
*p*	0.026	0.035
*F* (p)	78.08 (0.000)	36.87 (0.003)
(2) Drinking frequency during the past 30 days		
*b* (95% CI)	0.011 (0.001, 0.022)	0.013 (0.001, 0.025)
*p*	0.035	0.037
*F* (p)	45.26 (0.003)	10.55 (0.011)
(3) Weekly ethanol consumption (100 g/week)		
*b* (95% CI)	0.093 (0.014, 0.173)	0.107 (0.000, 0.213)
*p*	0.022	0.049
*F* (p)	31.60 (0.005)	23.73 (0.008)

MCI was defined as Mini-Cog [36]. Abbreviations: 95% CI represents 95% confidence interval. All regressions were adjusted for age, gender, ethnic minority, years of education, logarithm of household income, and province fixed effects. In particular, the MR model adjusted for the number of respondents’ parents who drink alcohol. The first-stage Cragg–Donald F statistics (with values of *p*) is test statistics of the weak instrument and overidentification tests.

## Data Availability

Not applicable.

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
