# Peer review of "Alcohol Consumption and Mild Cognitive Impairment: A Mendelian Randomization Study from Rural China"

_nutrients, 2022, doi:10.3390/nu14173596_

Round 1

Reviewer 1 Report

This study examines the causal effect of alcohol consumption on MCI in rural China. Although I have found the study interesting and timely, I have some major concerns to be addressed which I listed below:

- The introduction part is so long that the reader can get lost easily. I recommend it be rewritten in a more logical flaw and focused on the rationale of the study. 

- Please limit using conjuction words. It makes it hard to follow the manuscript. For example, on page 2, lines 41-44, the authors used the conjuction however, moreover and therefore consecutively. 

- Please give a reference for the following sentence: "However, there are few studies in Chinese rural areas".

- Please change this sentence "It should be the concerned the prevalence of MCI in Chinese rural areas." as follows "It should be concerned with the prevalence of MCI in Chinese rural areas."

- Please remove this from the introduction and add it to the methods section:  "MR is an ingenious causal research de-81 sign, which uses genes as instrumental variables [32,33]. The MR approach relies on random assignment of genes during meiosis in humans, resembling the random assignment into treatment groups in randomized controlled trials (RCT) [27,34-37]"

- Please add reference: "ALDH2 rs671, our genetic instrument, is shown to have the strongest association with alcohol consumption but don’t have association with MCI".

- The aim of the study should be rewritten. It is complicated and hard to follow as it is.

- Please merge Tables 1A and 1B and remove them to the results section. Same for Tables 2A and 2B. No need to give results in the methods section. The statistical analyses can be conducted to see whether there is a difference between the main characteristics (Table 1A and 1B).

- 1 decimal is enough for age and SD.

- Please add the definitions in tables 1A and 1B as a footnote. 

- Rewrite this sentence "By comparing Table 1A and Table 1B, it is can be seen that there was no significant difference in mainly characteristics between the total sample and the genetic sample, which 115 indicated that the genetic sample was representative of the total sample." as "By comparing Table 1A and Table 1B, it can be seen that there was no significant difference in main characteristics between the total sample and the genetic sample, which 115 indicated that the genetic sample was representative of the total sample."

- 2.3: measurement of alcohol consumption

- My major concern about the study is that it does not differ between the never-drinkers and previous drinkers and considers them as non-current drinkers. However, most of the previous drinkers stop drinking due to underlying health conditions. The authors should add a discussion part about this and discuss the limitations in detail.

- As indicated in the limitation, the majority of the participants were male. The authors can consider analysing only male participants as a sensitivity analysis. 

- Please change the total times of drinking to drinking frequency.

- Put the alcohol consumption results into the results section, not in the methods section.

- Page 5, line 155: Change is to has. Whether the participant has MCI..

- Page 8, line 180: Explanation of the drinking patterns?

- No need to provide columns in the results section. 

- "In order to prove the unbiased selection of genetic sample, we used multivariable linear regressions to analyze the genetic sample." This is a statistical analysis explanation, no need to give it in the results.

- Please rewrite: "Using ALDH2 rs671 as a validity genetic instrument variable relied on the critical assumption of relevance and the exclusion restriction"

- Please remove sentences that require references in the results section. The result section should only focus on the current study results, not what has been done previously. 

- Overall, the results section would benefit from rewriting in order to sound more scientific. For example, it would be better to remove the phrases such as "as can be seen" etc.

- The authors gave results for the different rural areas of China but did not discuss it further. If those results are not important, I would recommend removing them. 

- The authors claimed this is the first study, but please check this previous study and add it as a reference "Alcohol consumption and cognitive impairment in older men: a mendelian randomization study". It would be also better to add it to the introduction and discuss what this current study will add to the literature and what are the differences between this study and the previously published one. 

- Please remove the last sentence of the conclusion.

Reviewer 2 Report

Overall, this is an interesting piece of research with novel analysis. However, I have some minor comments. 

1. Alcohol consumption is not mentioned until the 4th paragraph (Line 55) in the introduction. This is strange as alcohol consumption is central to the hypothesis and is the first two words used in both the Title and Abstract. Please introduce alcohol at an earlier stage in the introduction

2. Line 95- I am unfamiliar with the term famers. Can you correct or else explain?

3. Line 114/115- you say it can be seen that there is no significant difference between characteristics but we can say for sure with visual inspection alone e.g. drinking time during past 30 days. It would be beneficial to include p-values. 

4. 131-134 I am quite confused about the ethanol calculations. The alcohol content of each of the types of alcohol vary considerably e.g. liquor could be 20-60%. Did you offer an average percentage of beer, wine and liquor in the questionnaire?

5. 231- different* not difference. 

6. 298-299- remove 'This section is not mandatory but can be 298 added to the manuscript if the discussion is unusually long or complex.'

Reviewer 3 Report

Review of manuscript entitled: “Alcohol Consumption and Mild Cognitive Impairment: A Mendelian Randomization Study from Rural China” authored by Yi Cui, Wei Si, Chen Zhu and Qiran Zhao

At the beginning I want to thank you for opportunity to review this interesting manuscript.

In the presented manuscript authors undertook a serious worldwide problem of alcohol intake and its effect on mild cognitive impairment. Introduction provides sufficient information about the problem, however in my opinion this paragraph is a little bit too long. Methods are described extensively but some improvements must be made like moving some parts to result section. Results are described nicely but Table 1A and 1B is very unclear and hard to interpret. Discussion is written logically and based on obtained results.

Overall manuscript is written good and touches important problem, however some improvements have to be made before final decision.

Major concerns:

  • Sample collection is described in detailed manner, although diagram representing sample collection process would be really helpful here
  • Did authors perform AUDIT or check medical history to verify if any of the participants is addicted to alcohol?
  • Table 1A and 1B – I do not quite understand manner of representing particular values here. It is very hard to interpret also. Moreover, I believe that this descriptive statistic of your dataset should be in the first paragraph of results section with simple description
    • Please provide exact values where it is possible to like sex (males & females), ethnic minority or income
    • Drinking times during past 30 days – does this mean inform us about how many days during that time people were drinking?
  • Table 2A and 2B – These are your results they should be placed in the proper paragraph (Results)

Minor concerns:

  • Lines 105-110 – authors repeated same information twice
  • Genotyping – authors stated that “DNA was extracted from saliva samples using the Illumina WeGene V2 Array.”, I could not find information about this array in the Web The question is just because of my curiosity, is it really used to isolate DNA from saliva or you put already isolated DNA on this array?

Round 2

Reviewer 3 Report

Authors responded to all my concerns. Thank you and I keep my fingers crossed for further evaluation of your work!